evolution, behaviour

coevolution, avian brood parasitism, egg colour, egg pattern, host defence, egg signatures

**Authors for correspondence:**
Eleanor M. Caves
e-mail: eleanor.caves@gmail.com
Claire N. Spottiswoode
e-mail: cns26@cam.ac.uk

†Deceased 17/11/2008

# Hosts elevate either within-clutch consistency or between-clutch distinctiveness of egg phenotypes in defence against brood parasites

Eleanor M. Caves[1,2], Tanmay Dixit[1], John F. R. Colebrook-Robjent[3,†], Lazaro Hamusikili[3], Martin Stevens[1,2], Rose Thorogood[1,4,5] and Claire N. Spottiswoode[1,6]

[1]Department of Zoology, University of Cambridge, Downing Street, Cambridge CB2 3EJ, UK
[2]Centre for Ecology and Conservation, University of Exeter, Penryn Campus, Penryn TR10 9FE, UK
[3]Musumanene Farm, Choma, Zambia
[4]HiLIFE Helsinki Institute of Life Sciences and [5]Research Programme in Organismal and Evolutionary Biology, Faculty of Biological and Environmental Sciences, University of Helsinki, Helsinki FI-00011, Finland
[6]FitzPatrick Institute of African Ornithology, DST-NRF Centre of Excellence, University of Cape Town, Rondebosch 7701, South Africa

EMC, 0000-0003-3497-5925; TD, 0000-0001-5604-7965; MS, 0000-0001-7768-3426; RT, 0000-0001-5010-2177; CNS, 0000-0003-3232-9559

In host–parasite arms races, hosts can evolve signatures of identity to enhance the detection of parasite mimics. In theory, signatures are most effective when within-individual variation is low ('consistency'), and between-individual variation is high ('distinctiveness'). However, empirical support for positive covariation in signature consistency and distinctiveness across species is mixed. Here, we attempt to resolve this puzzle by partitioning distinctiveness according to how it is achieved: (i) greater variation within each trait, contributing to elevated *'absolute* distinctiveness' or (ii) combining phenotypic traits in unpredictable combinations ('*combinatorial* distinctiveness'). We tested how consistency covaries with each type of distinctiveness by measuring variation in egg colour and pattern in two African bird families (Cisticolidae and Ploceidae) that experience mimetic brood parasitism. Contrary to predictions, parasitized species, but not unparasitized species, exhibited a negative relationship between consistency and combinatorial distinctiveness. Moreover, regardless of parasitism status, consistency was negatively correlated with absolute distinctiveness across species. Together, these results suggest that (i) selection from parasites acts on how traits combine rather than absolute variation in traits, (ii) consistency and distinctiveness are alternative rather than complementary elements of signatures and (iii) mechanistic constraints may explain the negative relationship between consistency and absolute distinctiveness across species.

## 1. Introduction

Whenever antagonistic coevolution involves mimicry as an offence, individuals are under selection to improve their detection of enemies [1,2]. In host–parasite arms races, this can result in hosts evolving 'signatures' of identity, which are individually distinctive phenotypes that facilitate detection of a parasite imposter while minimizing error [1,3]. Signatures are expected to be most effective if (i) host individuals vary *distinctively* with respect to other host signatures in a population, yet (ii) their individual signatures are *consistent* and vary as little as possible. Together, these adaptations are hypothesized to make it difficult for parasites to fool hosts because consistent signatures require mimics to be

an even better match, but distinctive signatures mean parasites can only mimic a small number of host phenotypes [1,3–6]. However, the questions of whether both adaptations co-occur within single systems, and how they manifest, have largely remained untested (but see [6–8]).

Avian brood parasitism, in which some brood-parasitic birds lay eggs that visually mimic those of their hosts to trick hosts into accepting the foreign egg as one of their own [9], provides a model system for exploring these questions. Over a century ago, Charles Swynnerton conducted a pioneering study of avian egg rejection in south-eastern Africa [3]. He hypothesized that if individual host females in a population were more phenotypically distinct from one another (adaptation (i) above), it would be harder for a parasite to lay eggs sufficiently mimetic to be accepted by a large proportion of host females, thus reducing Type II errors (false negatives; accepting a parasite egg). Subsequently, adaptation (ii) has also been applied to brood parasitism: if, within a clutch, females lay eggs that appear very similar to one another (i.e. more phenotypically consistent within individuals), it should be easier for females to identify parasitic eggs (lowering Type II errors [10]). Consistency within clutches should also have an added benefit by helping hosts to avoid erroneously rejecting their own eggs (thus also reducing Type I errors [11]). These hypotheses have generated the predictions that across species or populations, selection from brood parasites should be associated with both (i) greater interclutch variation (increased 'distinctiveness') between clutches laid by different females and (ii) reduced intraclutch variation (increased 'consistency') between eggs in a clutch laid by the same female, with respect to egg colour and pattern [12]. Here onwards, we use the terms 'distinctiveness' and 'consistency' (rather than 'interclutch variation' and 'intraclutch variation') because they are more generalizable to signature systems outside of avian brood parasitism, and because predictions are easier to visualize when selection for defence predicts elevation of both adaptations (rather than predicting elevation of one adaptation but reduction of the other, as for interclutch and intraclutch variation).

Despite the long-standing predictions that selection from parasitism should be associated with both elevated distinctiveness and consistency in host egg clutches, evidence supporting them is so far mixed. Studies relating to egg signature evolution have typically inferred the strength of selection from brood parasites from either the incidence of parasitism, or the intensity of host egg rejection. According to the predictions above, greater estimated selection from parasites should be correlated with increased distinctiveness and increased consistency. When testing these predictions, some studies have found that estimates of parasitism were positively correlated with distinctiveness across species [13–16], but others have not [4,17]. The comparative evidence for an association between parasitism and consistency is similarly mixed, with some support for a positive association [13,17–20] (but see for example [14,21]). Many studies within single species have also failed to find that consistency is associated with improved egg rejection ability [22–26] (but see [17,27,28]). However, compelling evidence for an association between parasitism and both distinctiveness and consistency has come from two studies of introduced populations, where hosts released from parasitism exhibited reduced distinctiveness and reduced consistency over

100–200 years of subsequent evolution [6–8]. How, then, can we reconcile these contrasting results?

One reason why studies have yielded conflicting results may be that distinctiveness and consistency need not occur hand-in-hand as defences against parasites. Rather, they may function as alternatives [29–31], if selection on one defence reduces the frequency of successful parasitism and thus weakens selection on the other [29]. Under such a 'strategy-blocking' scenario [32], hosts with high interclutch distinctiveness will rarely encounter a good match by the parasite, so hosts will not experience strong selection to further refine their detection and rejection via intraclutch consistency [12]. Reciprocally, hosts with high intraclutch consistency will often be able to detect a parasite even if it is a good match, and so not experience strong selection for interclutch distinctiveness. When comparing across host species, this strategy-blocking hypothesis generates different predictions to the traditional hypothesis. The latter predicts that consistency *and* distinctiveness in egg appearance should both be elevated, and positively correlated, in parasitized species, but only weakly correlated or uncorrelated across unparasitized species. By contrast, the strategy-blocking hypothesis predicts that either consistency *or* distinctiveness should be elevated in parasitized species. Therefore, there should be a negative correlation between consistency and distinctiveness in parasitized, but not unparasitized, species.

A second reason for the conflicting results may relate to how distinctiveness is achieved. Two key traits of bird eggs include colour and pattern, which can be quantified using a variety of metrics (e.g. hue, saturation, marking size, etc.). Typically, distinctiveness is quantified by summing the absolute level of variation within each metric ('absolute distinctiveness'). Higher absolute distinctiveness is conferred by greater interclutch variation in individual colour and pattern metrics (figure 1, left column). However, higher distinctiveness can arise in a second way if colour and pattern trait values co-occur in unpredictable combinations ('combinatorial distinctiveness'; figure 1, top row). In a species with high combinatorial distinctiveness, absolute levels of variation within each metric may be low, but because each individual's eggs comprise a different combination of values for each metric, they are all distinct from those of other individuals (figure 1*c*). However, combinatorial distinctiveness is rarely investigated, or separated from absolute distinctiveness.

We previously found that the eggs of host species had consistently elevated combinatorial distinctiveness (reflected by the correlation component of entropy), but not absolute distinctiveness (reflected by the variance component of entropy), compared with the eggs of non-hosts [33]. This suggested that selection from brood parasites may act on how hosts deploy phenotypic variation into combinations of trait values, rather than on absolute levels of variation within each metric. If selection acts more strongly on combinatorial than absolute distinctiveness, then this may account for the inconsistent findings of previous studies that have focussed only on absolute distinctiveness. Additionally, examining how consistency relates to each type of distinctiveness in both parasitized and unparasitized species may help clarify how selection from brood parasites acts on each defence.

Here we tested the predictions of the two hypotheses outlined above for the evolution of host egg signatures as a defence against brood parasites, by examining whether, and how, distinctiveness and consistency covary. The traditional

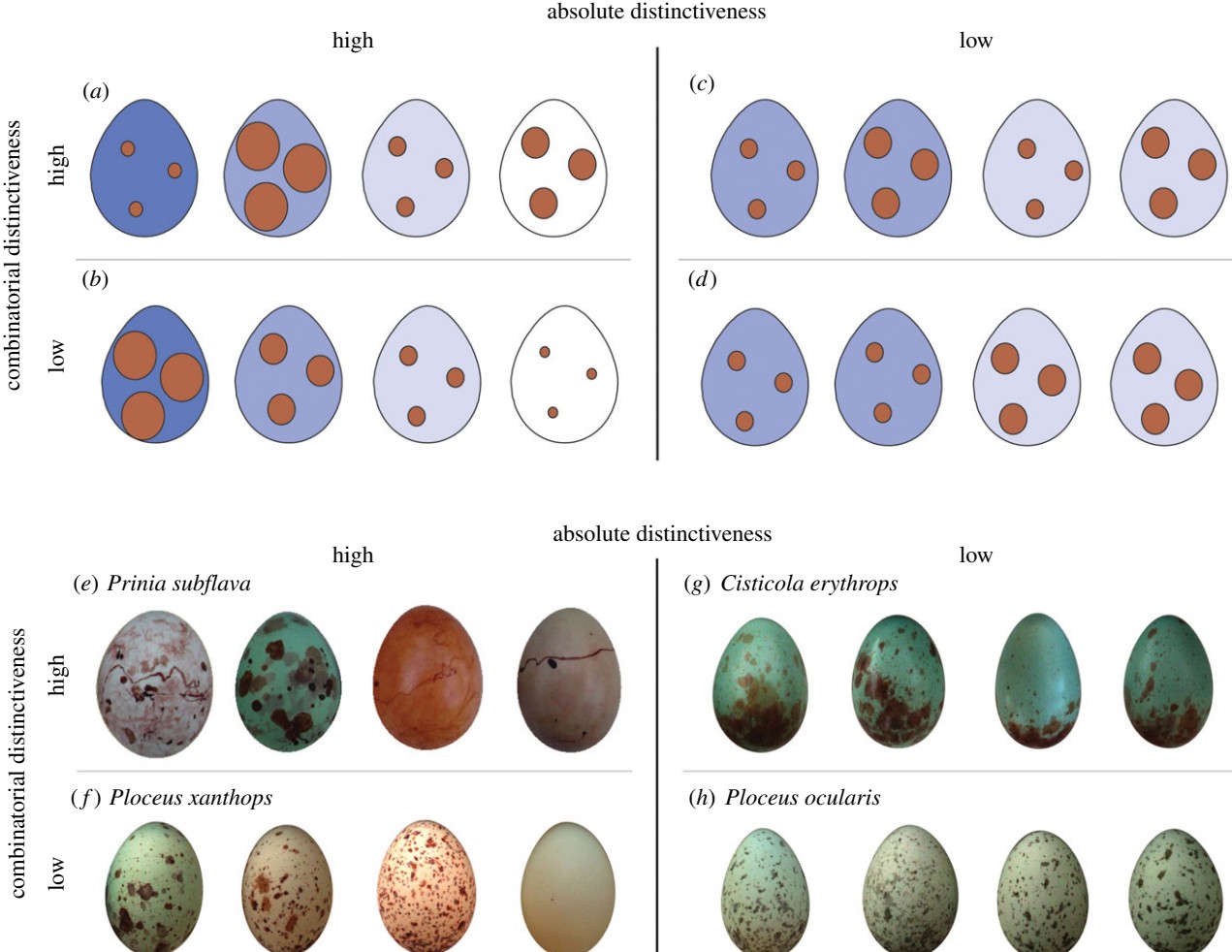

**Figure 1.** An illustration (*a–d*) and examples (*e–h*) of eggs from species with high and low levels of absolute and combinatorial distinctiveness. In each panel, the four eggs are from four different females. In the schematic (*a–d*), eggs vary in two traits (colour and pattern); for simplicity, each is described here by one metric (background hue and spot size). (*a*) The best scenario (for a host) is high variation in the metrics of spot size and background hue compared to other clutches (high absolute distinctiveness), and for the two to be uncorrelated (high combinatorial distinctiveness), making these eggs hardest for a parasite to mimic. (*b*) When spot size and background hue are correlated, one can be predicted from the other, so combinatorial distinctiveness is low, a situation which is not ideal for a host as any given parasite phenotype might be a good enough match to a larger subset of host phenotypes. (*c*) A species constrained for whatever reason to have low absolute distinctiveness is able to maximize variation among females by having high combinatorial distinctiveness. (*d*) The worst scenario (for a host) is low variation in both spot size and background hue and for the two to also be correlated, such that one can be predicted from the other. (*e–h*) Eggs from representative species in our dataset that exhibit high or low levels of absolute and combinatorial distinctiveness, relative to other species in our dataset.  (Online version in colour.)

hypothesis predicts that both consistency and distinctiveness are elevated, and positively correlated, across host species. The strategy-blocking hypothesis predicts that either consistency or distinctiveness should be elevated, and negatively correlated with one another, across parasitized species. We studied the same two families of African birds that inspired Swynnerton a century ago: the African warblers (Cisticolidae), many of which are parasitized by the cuckoo finch *Anomalospiza imberbis* [34]; and the weavers (Ploceidae), many of which are parasitized by the diederik cuckoo *Chrysococcyx caprius* [35]. These two systems have evolved independently from each other and have an ancient history of parasitism [36,37]. They are also notable for their diversity of egg phenotypes, with some species showing very high levels of interclutch distinctiveness resulting in elaborate 'signatures' of colour and pattern, and others not [33,38] (figure 2). Although each parasitic species has distinct host-races that mimic the variation in egg colour and pattern of their specialist host [35,38], parasitic females lay eggs haphazardly among nests of their host species

rather than targeting individual females with phenotypes that match their own [39,40]. Therefore, distinctiveness (whether achieved by absolute and/or combinatorial mechanisms) could be important for anti-parasitic defence in both systems.

## 2. Material and methods

### (a) Study system

We analysed 806 clutches (comprising 1942 eggs) from 11 warbler species (five parasitized, six unparasitized at our study site) and 14 weaver species (10 parasitized, four unparasitized at our study site) (electronic supplementary material, table S1). All eggs came from the Choma region of southern Zambia (near 16°47′S, 26°50′E) where they were collected by JFRCR and LH during the 1970s–1990s. We only analysed eggs collected in the Choma District and the districts of Monze and Mazabuka (centred 80 km and 130 km north-east of Choma, respectively). Data from this same set of eggs have previously been analysed as reported in [33,38].

*Proc. R. Soc. B* **288**: 20210326

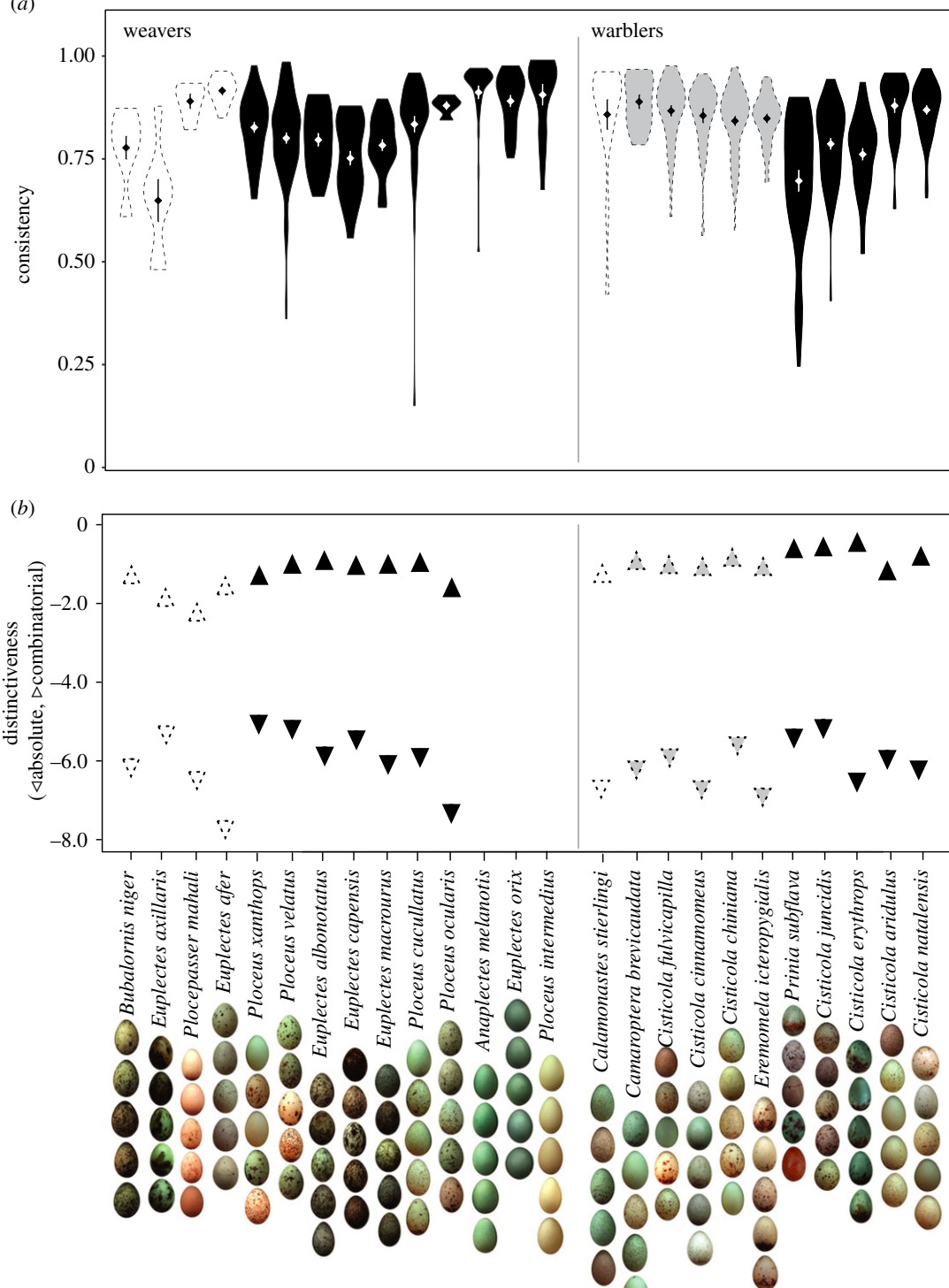

**Figure 2.** (*a*) Intraclutch consistency and (*b*) absolute (downward triangles) and combinatorial (upward triangles) distinctiveness for species of weaver (i) and warbler (ii). Black symbols indicate species currently parasitized at our study site. Dashed symbols indicate species not currently parasitized at our study site: light-grey dashed symbols indicate species with parasitism records from elsewhere in Africa; white dashed symbols indicate species with no parasitism records. Violin plots show distributions and ranges, diamonds show means, and bars show standard error. Consistency is a clutch-level measure, whereas distinctiveness is a species-level measure, so ranges are provided for consistency but not distinctiveness. Egg photos are representative examples of egg phenotypes, each from a different clutch. It is not possible to calculate entropy, and thus absolute and combinatorial distinctiveness, for the three species that lay only immaculate eggs (see Material and Methods); therefore, no values are displayed for those three species. (Online version in colour.)

To classify species as parasitized or unparasitized, we used data on 1490 collected clutches (range: 10–227 clutches, mean: 59.6 clutches) for our 25 study species, collected in the Choma region (including the Monze and Mazabuka districts) over 38 years by J.F.R.C.-R. and L.H. (electronic supplementary material, table S1). Each collected clutch was labelled as either parasitized or unparasitized, from which we calculated a parasitism rate in

the Choma region (range: 2.17–42.7%, electronic supplementary material, table S1). These rates are an imperfect index of parasitism pressure (e.g. some parasitic eggs could have been rejected before nests were found), but give a reliable indication of whether a species was regularly parasitized during the study period. We therefore categorized all species with non-zero parasitism rates as 'parasitized'. However, we cannot know with certainty whether

any currently unparasitized species have previously acted as hosts. As a precaution, we therefore repeated all analyses treating as parasitized five locally unparasitized species with published parasitism records from elsewhere in their range [41] (electronic supplementary material, table S1); this left only one unparasitized warbler in our dataset, reducing statistical power.

## (b) Quantifying egg colour and pattern

We focused on two traits which vary intraspecifically in the species in this study: colour and pattern. We quantified ten metrics following previously published methods [33] that included four measures of egg colour, one of egg luminance and five of egg pattern. In brief, to describe variation in colour, we used reflectance spectra to calculate avian photon catches (using data from the blue tit Cyanistes caeruleus [42]) for the UV, SW, MW, LW and double cones as measures of colour and luminance, respectively (as in [40]). To describe variation in pattern, we applied a granularity analysis [43] to digital photographs of eggs to quantify five pattern metrics, that have been previously applied to eggs [29,33,38,40,44]. These were (i) size of the predominant marking, (ii) contribution of the main marking to the overall pattern, (iii) contrast between pattern markings and the background, (iv) proportion of the egg covered by markings and (v) how dispersed markings were between the poles of the egg. Each pattern attribute and luminance was standardized for analyses by expressing it as a proportion of its maximum value within each family (warblers or weavers), so that the scale was comparable between all of the phenotypic attributes (as in [29,33]). Cone catch values were standardized to remove variation in absolute brightness, such that the standardized cone catch values sum to one (as in [33]).

As most of these metrics predict egg discrimination in cuckoo finch host species [29], we assumed that they represent biologically relevant metrics on which selection may act. We then used these metrics to calculate consistency, absolute distinctiveness and combinatorial distinctiveness in colour and pattern for each species, which we analysed in relation to current selection from brood parasites.

## (c) Quantifying consistency using multi-dimensional phenotypic space

To quantify phenotypic consistency within clutches, we calculated an index of overall diversity for our 10 focal colour and pattern metrics (four standardized single cone catches, luminance and five pattern metrics) combined. Following [29], we used a multi-dimensional phenotypic space (MDPS) analysis, in which each egg was mapped as a point in 10-dimensional space. The Euclidean distance between two points in this 10-dimensional space then provided a single measure of overall phenotypic distance between any two eggs in a group.

Each group was the clutch of a given female, which ranged from one to five eggs. To make clutches of different sizes directly comparable, we first eliminated all one-egg clutches from our dataset (as consistency cannot be calculated for one-egg clutches; this resulted in 704 clutches comprising 1840 eggs) and then carried out analyses on an 'effective clutch' (following [7]), defined as two randomly selected eggs from each clutch. A species index of intraclutch variability was taken as the mean of all pairwise comparisons within the effective clutches of a given species; we then subtracted this from one to yield an index of intraclutch consistency. Thus, high consistency values indicate that the eggs within a clutch are highly similar to one another in colour and pattern.

We also quantified distinctiveness using an MDPS method, which we show in the electronic supplementary material is conceptually analogous to absolute distinctiveness, and highly correlated with it.

## (d) Quantifying absolute and combinatorial distinctiveness

As measures of absolute and combinatorial distinctiveness, we used published values from [33], which analysed the same set of host and non-host species as examined here (see [33] for detailed methods). Briefly, species-specific values of 'differential' entropy (an extension of Shannon entropy for continuous variables [45,46]) were calculated using the same 10 phenotypic metrics described above (see the electronic supplementary material of [33] for entropy formulae). Values of entropy depend on the sum of two components, the variance component (the sum of the absolute variation of each metric, 'absolute distinctiveness') and the correlation component (how values for each metric are assembled within individuals; 'combinatorial distinctiveness').

Entropy was then decomposed into contributions from the variance and correlation components, which were uncorrelated ($F_{1,20} = 1.83$, $R^2 = 0.04$, $p = 0.19$), and which here represent the 'absolute' and 'combinatorial' components of distinctiveness, respectively. One egg per clutch was randomly selected for calculations of absolute and combinatorial distinctiveness to avoid pseudoreplication. Three weaver species in our study lay only immaculate eggs (Anaplectes melanotis, Euplectes orix and Ploceus intermedius; figure 2), meaning that the values for the five pattern attributes in these species were zero. Although MDPS analyses are not confounded by zero values, entropy cannot be measured on a comparable scale when any of the ten phenotypic attributes has a value of zero. Therefore, while we were able to calculate consistency, we were not able to calculate absolute and combinatorial distinctiveness for these three species.

## (e) Statistical analyses

We used R [47] to calculate distances in multi-dimensional phenotypic space and to implement linear models (using the lm function). First, we examined how consistency and both absolute and combinatorial distinctiveness were influenced by various factors. In these models, either consistency, absolute distinctiveness or combinatorial distinctiveness was the response variable, and parasitism status (parasitized or unparasitized), family membership (warbler or weaver) and sample size were fixed effects; where these contributed significantly to the model, we report their effects. Note that the models in which absolute or combinatorial distinctiveness were predicted by parasitism status, family membership and sample size are similar to those that are described in [33]. They differ only in that here we make use of an updated phylogeny for phylogenetic least-squares (PGLS) analyses and account for differences in sample size by including sample size as a covariate, rather than weighting the linear models by sample size, because the assumption of heteroscedasticity was met in our models, and because model coefficients are easier to interpret when they themselves are not weighted. For comparison with the previous study, however, we repeated all linear models weighting by sample size, and results were largely consistent with those using sample size as a covariate (electronic supplementary material, tables S2, S3 and S4). In the main text, we report the results of analyses that include sample size as a covariate, for consistency among the formulation of linear models in this study.

Second, we asked whether there was a positive or a negative relationship between consistency and either absolute or combinatorial distinctiveness across species. We modelled consistency as the response variable, absolute or combinatorial distinctiveness as the predictor variable, and parasitism status, family membership, and sample size as covariates. Finally, we asked whether the relationships between consistency and either absolute or combinatorial distinctiveness differed for parasitized and unparasitized species, by including an interaction between the predictor variable and parasitism status in these models.

Sample sizes varied among species (electronic supplementary material, table S1), so we checked for any bias caused by unequal sample sizes bias in two ways. First, we included sample size as a covariate in analyses using the full dataset. Second, we resampled each species to the minimum sample size ($n = 5$) and repeated analyses using consistency recalculated using this smaller dataset, to check whether results were affected (none were; see electronic supplementary material, table S2). In models analysing resampled datasets, sample size was not included as a fixed effect. We used Cook's distance [48] to identify outliers in any analyses; where outliers were identified, we repeated analyses with those species excluded (electronic supplementary material, table S3). In some models, there were minor deviations from normality of residuals, so as a precaution we repeated all of the above analyses with the dependent variable expressed in ranks, but all conclusions were unchanged (electronic supplementary material, table S2).

## (f) Accounting for phylogenetic relatedness

We used the R package *phytools* [49] to calculate Pagel's $\lambda$ [50], a measure of phylogenetic signal that indicates the level of match between the model's residuals and the structure of the phylogeny. $\lambda$ normally varies between zero, indicating phylogenetic independence, and one, indicating direct covariance between the species' phenotypic values of interest and phylogenetic structure.

Some species in our study have either not been formally placed on a phylogenetic tree, or placed but with low confidence; we therefore used birdtree.org [51] to compile 100 trees with branch lengths for our focal species. We then calculated both $\lambda$ and a *p*-value for a log-likelihood test of significant phylogenetic signal and examined the mean and standard deviation of each across the 100 trees. We found no evidence of phylogenetic signal in intraclutch consistency ($\lambda < 0.001$ and $p = 1$ in all trees), absolute distinctiveness ($\lambda < 0.001$ and $p = 1$ in all trees) or combinatorial distinctiveness ($\lambda$ mean $\pm$ s.d. $= 0.29 \pm 0.03$; range of $p$ across all trees: 0.07–0.12). However, all analyses were repeated using PGLS models implemented by the R package *caper* [52], to account for the fact that related species are not statistically independent owing to shared phylogenetic history [53]. PGLS models did not include family membership as a fixed effect.

## 3. Results

## (a) Consistency, absolute distinctiveness and combinatorial distinctiveness in parasitized versus unparasitized species

First, we asked whether parasitized species have consistently higher levels of each defence than unparasitized species (figure 2). We found no significant differences in consistency between currently parasitized and unparasitized species (slope $\pm$ s.e. $= 0.004 \pm 0.033$, $t_{21} = 0.12$, $p = 0.91$). Results were similar when we took a more restrictive definition of parasitism status, scoring the five species parasitized outside of Choma (our main study area) as parasitized (slope $\pm$ s.e. $= -0.045 \pm 0.041$, $t_{21} = -1.08$, $p = 0.29$). Repeating the analyses above while taking phylogenetic structure into account (and therefore not modelling family membership as a factor) did not change any of the conclusions above: consistency did not differ in relation to parasitism status at our study site (slope $\pm$ s.e. $= 0.005 \pm 0.037$, $t_{16} = 0.15$, $p = 0.88$), or when species parasitized outside of Choma were treated as parasitized (slope $\pm$ s.e. $= -0.048 \pm 0.052$, $t_{16} = -0.92$, $p = 0.37$). We also repeated all analyses (i) with species-specific values for consistency calculated from data resampled to

the sample size of the least sampled species; (ii) omitting the three species that lay only immaculate eggs and (iii) with the dependent variable expressed in ranks, and in all cases conclusions were unchanged (electronic supplementary material, table S2).

According to a very similar previous analysis [33], differing only with respect to the exact formulation of the linear models used and in applying a more updated phylogenetic tree for PGLS analyses (see Material and Methods), we found that only combinatorial distinctiveness, and not absolute distinctiveness, was consistently elevated in parasitized over unparasitized species (combinatorial distinctiveness: slope $\pm$ s.e. $= -0.31 \pm 0.12$, $t_{18} = -2.67$, $p = 0.02$; absolute distinctiveness: slope $\pm$ s.e. $= -0.28 \pm 0.30$, $t_{18} = -0.94$, $p = 0.36$). These results were unchanged when performing the analyses using ranked data, with a phylogenetic correction, or when weighting the results by sample size (electronic supplementary material, table S4).

In summary, we found no relationship between parasitism status and either consistency or absolute distinctiveness in egg phenotype, contrary to the traditional pair of predictions that selection from brood parasites should result in higher levels of both defences. We did, however, find that combinatorial distinctiveness is elevated in parasitized over unparasitized species, replicating our previous finding [33].

## (b) The relationship between intraclutch consistency and both absolute and combinatorial distinctiveness

We then examined how consistency related to both absolute and combinatorial distinctiveness. In an additive model, the relationship between consistency and absolute distinctiveness was significantly negative (slope $\pm$ s.e. $= -0.06 \pm 0.02$, $t_{17} = -3.04$, $p = 0.007$). Allowing an interaction between parasitism status and absolute distinctiveness, the interaction term was not significant (slope $\pm$ s.e. $= -0.039 \pm 0.04$, $t_{16} = -0.97$, $p = 0.35$). The interaction term remained non-significant when locally unparasitized species were treated as parasitized; when excluding two species (*Euplectes axillaris* and *Plocepasser mahali*) that were identified as outliers using Cook's distance; when applying a phylogenetic correction; or when using values of consistency generated from five randomly sampled clutches per species (electronic supplementary material, table S3). Therefore, the relationship between consistency and absolute distinctiveness did not detectably differ between parasitized and unparasitized species.

In contrast with the negative relationship between consistency and absolute distinctiveness, we found no significant relationship between consistency and combinatorial distinctiveness across all species (slope $\pm$ s.e. $= -0.11 \pm 0.06$, $t_{17} = -1.79$, $p = 0.10$). However, in a model allowing an interaction between combinatorial distinctiveness and consistency, the interaction term was significant (slope $\pm$ s.e. $= 0.15 \pm 0.07$, $t_{16} = 2.05$, $p = 0.05$), showing that the slope of the relationship between consistency and combinatorial distinctiveness is different in parasitized versus unparasitized species (figure 3b). In particular, the relationship between consistency and combinatorial distinctiveness was significantly negative in a model including only parasitized species (slope $\pm$ s.e. $= -0.18 \pm 0.07$, $t_8 = -2.72$, $p = 0.03$), but not significant in a model including only unparasitized species (slope $\pm$

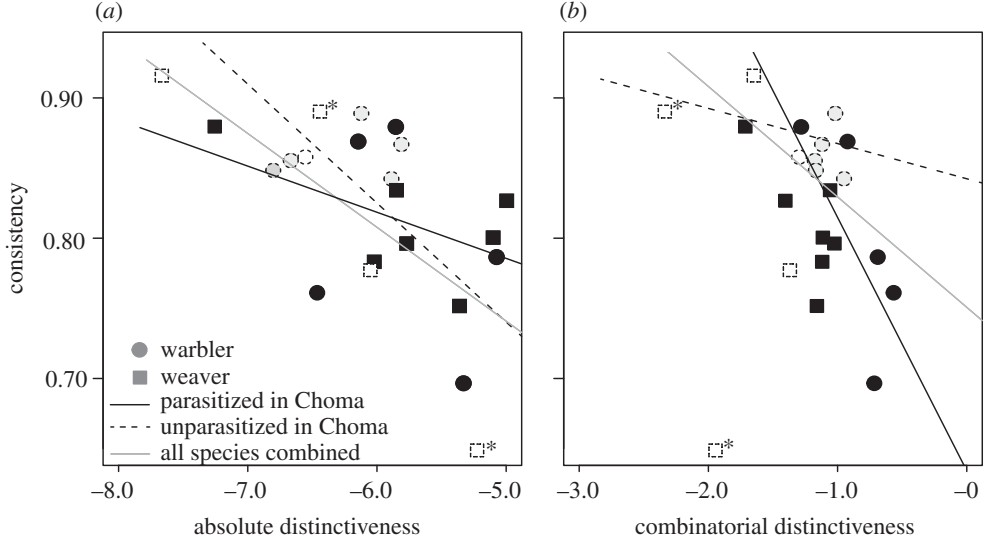

**Figure 3.** A visualization of the linear models describing the relationship between intraclutch consistency and (*a*) absolute distinctiveness and (*b*) combinatorial distinctiveness, while accounting for family membership and sample size, in warbler (circles) and weaver (squares) species. Black symbols and lines indicate species currently parasitized at our study site. Dashed symbols and lines indicate species not currently parasitized at our study site; light-grey dashed symbols indicate species with parasitism records from elsewhere in Africa, and white symbols indicate species with no parasitism records. The slopes for parasitized and unparasitized species significantly differ in (*b*) but not in (*a*) (see main text) and the line of best fit for all species combined (solid grey) is provided for comparison. The species identified as statistical outliers, *Plocepasser mahali* and *Euplectes axillaris*, have been marked with an asterisk. A visualization of the relationship between intraclutch consistency and both absolute and combinatorial distinctiveness, with 95% confidence intervals, is available in electronic supplementary material, figure S1.

s.e. = −0.03 ± 0.12, $t_6 = -0.21$, $p = 0.84$). The interaction between combinatorial distinctiveness and parasitism status became marginally non-significant when we incorporated a phylogenetic correction ($p = 0.07$) and used resampled data ($p = 0.07$), and was not significant when species parasitized elsewhere in Africa were treated as parasitized or when the outliers *P. mahali* and *E. axillaris* were removed (electronic supplementary material, table S3). However, as noted in the Material and Methods, these supplementary analyses had low power to distinguish between parasitized and unparasitized species due to low sample sizes.

## 4. Discussion

The expectation that hosts under selection from parasites should evolve signatures that are distinctive among females, yet consistent within a female, to help them recognize their own eggs, has received extensive testing in avian brood parasite-host systems. Here, in two families of African birds in which several species are respectively parasitized by a cuckoo and the cuckoo finch, we found support for two potential reasons why previous results have been inconsistent in their support for this elegant hypothesis. First, in parasitized species, we found a negative relationship between consistency and combinatorial distinctiveness, which supports the hypothesis that these are alternative rather than simultaneous defences. This finding goes against the traditional expectation that selection for effective signatures of identity should elevate both consistency and distinctiveness [5]. Second, we found a negative relationship between consistency and absolute distinctiveness that was the same for parasitized and unparasitized species, suggesting that selection from brood parasites favours combinatorial rather than absolute distinctiveness (i.e. distinctiveness achieved by assembling unpredictable combinations of egg colour and

pattern trait values, rather than higher levels of variation in each). Therefore, previous studies might have missed a relationship between consistency and distinctiveness depending upon how they defined and quantified distinctiveness. These results lead us to ask two questions: (i) why should hosts elevate only consistency or combinatorial distinctiveness rather than both? (ii) Why is the relationship between consistency and absolute distinctiveness negative in both hosts and non-hosts?

### (a) Why should hosts elevate only a single defense rather than both?

Our results support the hypothesis that elevating either consistency or combinatorial distinctiveness allows hosts to successfully detect parasitic eggs. Experimental data currently exist for two of our focal species to examine whether this is also supported when parasitic egg detection is measured directly. The tawny-flanked prinia *Prinia subflava* lays eggs with high absolute distinctiveness (figure 2), while the red-faced cisticola *Cisticola erythrops* lays only blotched or stippled eggs with a turquoise background that have low absolute distinctiveness (the lowest of any warbler in our dataset). At first sight, the two species appear to have very different defences: *P. subflava* eggs appear quite variable while *C. erythrops* eggs are not. Both species, however, have high combinatorial distinctiveness and low consistency, and both reject parasitic eggs equally well [29]. Thus, this behavioural evidence supports the idea that parasitism pressure acts on combinatorial distinctiveness, and that species with high combinatorial distinctiveness will not experience strong selection for consistency. Experimental data on more species, especially those that exhibited high consistency and low combinatorial distinctiveness here (e.g. desert cisticola *C. aridulus*), will help to reveal whether

hosts prioritizing either consistency or distinctiveness can have similar outcomes for the detection of parasitic eggs.

Prioritizing one defence over another also provides support for strategy-blocking, whereby selection on one defence lowers selection on another defence [32]. For example, species with high distinctiveness will rarely encounter a parasitic egg that is a good match for their own, so hosts will not experience strong selection to further refine either consistency or egg rejection behaviour. Interestingly, support for a strategy-blocking hypothesis was only found when distinctiveness was measured as combinatorial distinctiveness and not as absolute distinctiveness. Thus, precisely what type of individually identifying information is encoded in an egg phenotype may determine how consistency and distinctiveness are expressed by a given female or species. This means that the way in which distinctiveness is measured can affect the conclusions of studies on interclutch variation.

Why are some host species consistent in their egg phenotypes, and others (combinatorially) distinct? Which defence is elevated by coevolution with parasites may be influenced by selection from ecological factors aside from brood parasitism: certain egg phenotypes may be costly with respect to thermoregulation, protection from UV radiation, or camouflage [54–57], and increased susceptibility to host colonizations by other species or host races of the parasite [29,38], potentially limiting distinctiveness. Moreover, non-adaptive factors may also mean that the null hypothesis is not necessarily that both consistency and distinctiveness should be low in the absence of parasitism. For example, non-adaptive mechanisms might account for the high levels of consistency observed in some unparasitized species; we could speculate that certain egg phenotypes may be relatively invariant within females for mechanistic reasons during pigment deposition. Variation in such potential constraints may influence whether high consistency or distinctiveness is favoured, and thus why the closely related, sympatric species studied here have taken divergent trajectories from their similar phylogenetic and ecological starting points.

Finally, one inevitable problem shared by all comparative studies relating defensive traits to current selection from brood parasites is that coevolution is dynamic. Species that are not currently parasitized may have experienced selection from brood parasites in the past. For example, the rattling cisticola *Cisticola chiniana* is currently unparasitized at our study site yet shows intermediate levels of both interclutch distinctiveness and discrimination behaviour. As it is parasitized elsewhere, this suggests that it may have locally defeated its parasite [29]. Similarly, the grey-backed camaroptera *Camaroptera brevicaudata* is parasitized by emerald cuckoos *Chrysococcyx cupreus* elsewhere in its range [39] although no parasitism records exist from the Choma region despite intensive searching (by J.F.R.C.-R. and L.H.). Gene flow from emerald cuckoo-parasitized populations might help to account for its variously blue, white and immaculate to heavily blotched eggs. We attempted to reduce any such confounding effects by reanalysing our data with such species re-assigned and excluded. An undetected history of parasitism might also account for the surprisingly high levels of distinctiveness shown by some species with no records of parasitism anywhere in their range, such as the red-billed buffalo weaver *Bubalornis niger* and fan-tailed widowbird *Euplectes axillaris*.

## (b) Why is the relationship between consistency and absolute distinctiveness negative in both hosts and non-hosts?

We found a negative relationship between consistency and absolute distinctiveness for both parasitized and unparasitized species, suggesting that selection from brood parasites cannot entirely explain these results. One possibility is that this negative relationship arises from a mechanistic constraint that is not directly related to parasitism. Instead, it may be indicative of a fundamental trade-off between mechanisms which produce repeatability in signatures (which can increase consistency within clutches) and mechanisms which produce randomness in signatures (which can promote distinctiveness between the clutches of different females). Repeatability and randomness are somewhat antithetical, as repeatability requires predictability, whereas randomness entails unpredictability [58]. Very little is known about the mechanisms of colour and pattern generation in the shell gland [30,59]. However, we can speculate that consistency within clutches must be produced by mechanisms which ensure repeatability in pattern, whereas distinctiveness could be produced through some level of randomness in the generation of signatures. If this is the case, the mechanisms involved in producing consistency may reduce distinctiveness and vice versa. This could potentially explain why we find a negative relationship in both parasitized species (where we would expect selection on consistency and distinctiveness) and unparasitized species (where we would expect no or weaker selection).

## 5. Conclusion

Overall, our data suggest that in these two bird families, the egg signatures of different species lie on a spectrum of low distinctiveness and high consistency to high distinctiveness and low consistency. When distinctiveness is generated specifically by combinatorial information, this spectrum is seen only in parasitized species, consistent with a strategy-blocking scenario whereby selection for one defence reduces the strength of selection favouring the other defence. Previous studies found varying levels of support for the hypotheses that selection from parasitism should elevate consistency and/or distinctiveness in host eggs. Our results suggest that this could be explained if those host families also rely primarily on combinatorial rather than absolute distinctiveness in defence against brood parasites. We suggest that using entropy as a way of conceptualizing and measuring both combinatorial and absolute distinctiveness may help to clarify the relationship between distinctiveness and either incidence of parasitism or rejection ability.

This study is correlative, but comparative analyses together with behavioural and physiological experiments on a wider range of species will help to test the generality of the mechanisms proposed here. In particular, we should investigate the mechanisms involved in producing consistent or distinctive eggs, and explore the costs of these phenotypes from other sources of selection besides brood parasitism. Future research should also test whether negative relationships between consistency and measures of distinctiveness are a default property of birds' eggs, or indeed biological patterns in general, rather than an adaptation to parasitism. More broadly, by considering the relationship between consistency and distinctiveness across a

range of systems involving discrimination of self from non-self, we may enhance our understanding of how and why these systems follow a variety of trajectories when hosts coevolve with mimetic antagonists.

Data accessibility. Data and R codes associated with this manuscript are available from the Dryad Digital Repository https://doi.org/10.5061/dryad.02v6wwq34 [60].

Authors' contributions. C.N.S., M.S. and E.M.C. conceived the study; J.F.R.C.-R. and L.H. collected eggs and breeding data and helped inspire the study; E.M.C. collected the data from J.F.R.C.-R.'s egg collection; M.S. and E.M.C. performed visual modelling; E.M.C. analysed the data; E.M.C., T.D., C.N.S., M.S. and R.T. interpreted the data and E.M.C., T.D., C.N.S. and R.T. wrote the manuscript, with contributions from M.S.

Competing interests. We declare we have no competing interests.

Funding. E.M.C. was supported by the Pomona College-Downing College Student Exchange Scholarship; T.D. by a Balfour Studentship from the Department of Zoology, University of Cambridge; R.T. by an Independent Research Fellowship from the Natural Environment Research Council UK (grant no. NE/K00929X/1) and a start-up grant from the Helsinki Institute of Life Science (HiLIFE), University of Helsinki; MS by a BBSRC David Phillips Fellowship (grant no. BB/G022887/1), and C.N.S. by a Royal Society Dorothy Hodgkin Fellowship and BBSRC David Phillips Fellowship (grant no. BB/J014109/1).

Acknowledgements. In Zambia, we thank Emma and Ian Bruce-Miller for their hospitality, Jeroen Koorevaar for help with egg photography, and all of J.F.R.C.-R.'s many field assistants who found the nests involved in this study. We thank Jess Lund and two anonymous reviewers for helpful comments on earlier drafts. We thank Dr Edwin Iversen for helping to devise the application of entropy to egg signatures.

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
