## [Peer Review File · Proceedings of the Royal Society B: Biological Sciences]

Review History

RSPB-2021-0326.R0 (Original submission)

Review form: Reviewer 1

Recommendation

Accept with minor revision (please list in comments)

Scientific importance: Is the manuscript an original and important contribution to its field?

Excellent

General interest: Is the paper of sufficient general interest?

Excellent

Quality of the paper: Is the overall quality of the paper suitable?

Excellent

Is the length of the paper justified?

Yes

Should the paper be seen by a specialist statistical reviewer?

No

Do you have any concerns about statistical analyses in this paper? If so, please specify them explicitly in your report.

No

It is a condition of publication that authors make their supporting data, code and materials available - either as supplementary material or hosted in an external repository. Please rate, if applicable, the supporting data on the following criteria.

Is it accessible?

Yes

Is it clear?

Yes

Is it adequate?

Yes

Do you have any ethical concerns with this paper?

No

Comments to the Author

It has been hypothesized that hosts of brood parasites should minimize intraclutch egg variation (i.e., “consistency”) and maximize interclutch egg variation (“distinctiveness”) within a species to circumvent the evolution of egg mimicry by brood parasites. To date, studies investigating this possibility have lacked consensus. The authors of this manuscript investigated possible explanations for these inconsistencies by focusing on the eggs of two ancient host lineages in Africa, weavers and warblers, which are parasitized by parasites that lay mimetic eggs. In addition to testing the traditional hypothesis that both consistency and distinctiveness should be elevated, they tested the strategy-blocking hypothesis which suggests that either consistency or distinctiveness can be elevated to thwart parasitism, rather than requiring both simultaneously.

The authors found evidence that these strategies can be alternatives to combat parasitism rather than requiring both to be expressed simultaneously. Contrary to expectations, combinatorial distinctiveness was more important than absolute distinctiveness. Importantly, they also found a negative relationship between consistency and absolute distinctiveness in hosts and non-hosts alike, indicating that not surprisingly, other factors affect egg appearance (e.g., Wisocki et al. 2020). This is an exciting and novel study building on the results of their previous work and I have relatively few comments.

That only consistency or distinctiveness is required to combat parasitism is similar to other defenses (e.g., egg rejection and aggressive nest defense) where many hosts, especially those of the cowbird, invest most of their effort in a single defense. In the Introduction you mention Lahti’s work is one of the rare cases where we see low intraclutch and high interclutch variation in eggs of a host species. Given your findings that this combination appears to be unnecessary, why is that situation unique? If only consistency or distinctiveness is necessary, why are both present in this system? It would seem costly to invest in both if it’s not required.

Review form: Reviewer 2

Recommendation

Accept with minor revision (please list in comments)

Scientific importance: Is the manuscript an original and important contribution to its field?
Good

General interest: Is the paper of sufficient general interest?
Excellent

Quality of the paper: Is the overall quality of the paper suitable?
Excellent

Is the length of the paper justified?
Yes

Should the paper be seen by a specialist statistical reviewer?
No

Do you have any concerns about statistical analyses in this paper? If so, please specify them explicitly in your report.
No

It is a condition of publication that authors make their supporting data, code and materials available - either as supplementary material or hosted in an external repository. Please rate, if applicable, the supporting data on the following criteria.

Is it accessible?
Yes

Is it clear?
Yes

Is it adequate?
Yes

Do you have any ethical concerns with this paper?
No

Comments to the Author

I found this to be a fascinating study, very well written with a clear thoughtful framework that captures the nuances associated with defensive adaptations to brood parasitism.

I only have a couple of general comments. The first is about the categorical assignment of whether a host species is parasitized or not. This is based on nest records in the area from which the clutches were collected which is great, but I would like to see more details about exactly how these categories were assigned. There were 1457 nest records for 25 species – so an average about 60 nests per species. Surely some of these species had many more nest records and some had less? Was there much variation in the rate of parasitism in the species that were identified as hosts? If so, does that provide any predictive power regarding egg signature variation? In sum, I thought this aspect was somewhat glossed over and I would certainly appreciate more details about it. Thinking of being parasitized as a yes or no trait seems slightly simplistic to me when it seems that it is more likely a continuum.

Perhaps a table with breeding records per species, number of clutches parasitized and frequency of clutches parasitized would be useful? Possibly a frequency histogram of frequency of parasitism – if it is clearly bimodal then that would justify the categorization of either yes or no for species as being parasitized

The other aspect I struggled to wrap my head around is the basic argument that that selection from brood parasites should result in higher levels of both defenses. Thus consistency and combinatorial distinctiveness as both potential adaptations against parasitism are basically

predicted to be higher in parasitized species? But if either of these are effective adaptations against parasitism – shouldn't it be associated with reduced parasitism? So if you classify species as parasitized or not, I would have predicted that the parasitized species would have less consistency and combinatorial distinctiveness, thereby making them more susceptible to getting fooled by the parasites. Is the above argument a possible reason why “previous results have been inconsistent in their support for this elegant hypothesis” line 408

Decision letter (RSPB-2021-0326.R0)

04-May-2021

Dear Dr Caves

I am pleased to inform you that your manuscript RSPB-2021-0326 entitled "Hosts elevate either within-clutch consistency or between-clutch distinctiveness of egg phenotypes in defence against brood parasites" has been accepted for publication in Proceedings B.

The referee(s) have recommended publication, but also suggest some minor revisions to your manuscript. Therefore, I invite you to respond to the referee(s)' comments and revise your manuscript. Because the schedule for publication is very tight, it is a condition of publication that you submit the revised version of your manuscript within 7 days. If you do not think you will be able to meet this date please let us know.

It is a condition of publication that data supporting your paper are made available either in the electronic supplementary material or through an appropriate repository. Please see our Data Sharing Policies <https://royalsociety.org/journals/authors/author-guidelines/#data>.

Sincerely,

Dr Maurine Neiman

Associate Editor

Board Member: 1

Comments to Author:

This exceptionally well-written paper presents data on egg phenotype in several bird species in Africa and asks whether heterospecific brood parasitism has selected for increased intraclutch consistency or interclutch distinctiveness (or neither). The two reviewers and I were impressed by the quality of the manuscript -- the evolutionary hypotheses, methods, and results are all well explained -- and we found the results exciting and novel. The two reviewers each have several comments which should improve the paper even further. My own comments are minor:

First, although non-adaptive hypotheses are mentioned in the discussion, I think they deserve a bit more attention throughout the paper, possibly even in the introduction. There are many aspects of egg phenotype that seem to be relatively invariant for individual females, even when variation might be adaptive (notably, egg size). The proximate physiological mechanisms necessary to alter, say, the degree of shell pigmentation between subsequent eggs in a clutch may be formidable. Or females might be constrained in the amounts of biliverdin or other pigments, and so forth. Alternatively, there may be no selective advantage or disadvantage to consistency, so it may be essentially random. I mention this partly because it might help interpret the results, but also because it is important to keep the null hypothesis in mind. In the discussion (line 475), for example, you mention that some unparasitized species show a "surprisingly high" level of interclutch distinctiveness (surprising because they are not parasitized). But perhaps this simply indicates that there is no selection *against* distinctiveness, so it reflects variation in individual condition (or something else about individual phenotype).

Second, I appreciated the schematic figure 1, but I think it would be even better if you had shown photos of your eggs with real examples of consistency and distinctiveness. I realize that this may be impossible/too much work, so it's only a suggestion if feasible.

Finally, this may need just a quick clarification, but I was a little concerned about the inclusion of some species that lay immaculate eggs. It seems that including these could throw off statistical analyses that are otherwise dealing with the extent of color/patterning, even if the methods themselves can theoretically accommodate such data. For example, do the immaculate eggs vary as much in background color as the patterned eggs do, or are they more uniform overall (not within vs between clutches, but at the population level)?

I enjoyed reading this exciting paper and hope that these comments are helpful.

Reviewer(s)' Comments to Author:

Referee: 1

Comments to the Author(s)

It has been hypothesized that hosts of brood parasites should minimize intraclutch egg variation (i.e., "consistency") and maximize interclutch egg variation ("distinctiveness") within a species to circumvent the evolution of egg mimicry by brood parasites. To date, studies investigating this possibility have lacked consensus. The authors of this manuscript investigated possible explanations for these inconsistencies by focusing on the eggs of two ancient host lineages in Africa, weavers and warblers, which are parasitized by parasites that lay mimetic eggs. In addition to testing the traditional hypothesis that both consistency and distinctiveness should be elevated, they tested the strategy-blocking hypothesis which suggests that either consistency or distinctiveness can be elevated to thwart parasitism, rather than requiring both simultaneously.

The authors found evidence that these strategies can be alternatives to combat parasitism rather requiring both to be expressed simultaneously. Contrary to expectations, combinatorial distinctiveness was more important than absolute distinctiveness. Importantly, they also found a negative relationship between consistency and absolute distinctiveness in hosts and non-hosts alike, indicating that not surprisingly, other factors affect egg appearance (e.g., Wisocki et al. 2020). This is an exciting and novel study building on the results of their previous work and I have relatively few comments.

That only consistency or distinctiveness is required to combat parasitism is similar to other defenses (e.g., egg rejection and aggressive nest defense) where many hosts, especially those of the cowbird, invest most of their effort in a single defense. In the Introduction you mention Lahti's work is one of the rare cases where we see low intraclutch and high interclutch variation in eggs of a host species. Given your findings that this combination appears to be unnecessary, why is that situation unique? If only consistency or distinctiveness is necessary, why are both present in this system? It would seem costly to invest in both if it's not required.

Referee: 2

Comments to the Author(s)

I found this to be a fascinating study, very well written with a clear thoughtful framework that captures the nuances associated with defensive adaptations to brood parasitism.

I only have a couple of general comments. The first is about the categorical assignment of whether a host species is parasitized or not. This is based on nest records in the area from which the clutches were collected which is great, but I would like to see more details about exactly how these categories were assigned. There were 1457 nest records for 25 species – so an average about 60 nests per species. Surely some of these species had many more nest records and some had less? Was there much variation in the rate of parasitism in the species that were identified as hosts? If so, does that provide any predictive power regarding egg signature variation? In sum, I thought this aspect was somewhat glossed over and I would certainly appreciate more details about it. Thinking of being parasitized as a yes or no trait seems slightly simplistic to me when it seems that it is more likely a continuum.

Perhaps a table with breeding records per species, number of clutches parasitized and frequency of clutches parasitized would be useful? Possibly a frequency histogram of frequency of parasitism – if it is clearly bimodal then that would justify the categorization of either yes or no for species as being parasitized

The other aspect I struggled to wrap my head around is the basic argument that that selection from brood parasites should result in higher levels of both defenses. Thus consistency and combinatorial distinctiveness as both potential adaptations against parasitism are basically predicted to be higher in parasitized species? But if either of these are effective adaptations against parasitism – shouldn't it be associated with reduced parasitism? So if you classify species as parasitized or not, I would have predicted that the parasitized species would have less consistency and combinatorial distinctiveness, thereby making them more susceptible to getting fooled by the parasites. Is the above argument a possible reason why “previous results have been inconsistent in their support for this elegant hypothesis” line 408

Author's Response to Decision Letter for (RSPB-2021-0326.R0)

See Appendix A.

Decision letter (RSPB-2021-0326.R1)

01-Jun-2021

Dear Dr Caves

I am pleased to inform you that your manuscript entitled "Hosts elevate either within-clutch consistency or between-clutch distinctiveness of egg phenotypes in defence against brood parasites" has been accepted for publication in Proceedings B.

Data Accessibility section

Open Access

Paper charges

Sincerely,

Dr Maurine Neiman

Associate Editor:

Board Member

Comments to Author:

Thanks to the authors for addressing the minor comments raised by myself and the two reviewers in the previous review. This is a beautifully written and interesting manuscript.

Appendix A

Manuscript ID RSPB-2021-0326: Response to Reviewers

Associate Editor

Comments to Author:

This exceptionally well-written paper presents data on egg phenotype in several bird species in Africa and asks whether heterospecific brood parasitism has selected for increased intraclutch consistency or interclutch distinctiveness (or neither). The two reviewers and I were impressed by the quality of the manuscript -- the evolutionary hypotheses, methods, and results are all well explained -- and we found the results exciting and novel.

- **Reply:** We thank the editor and the reviewers for their positive assessment of our manuscript, which we were delighted to receive, as well as their constructive comments and feedback. We have provided detailed responses to each comment below.
- We have also made one change that was not requested by the reviewers. Previously, in Figure 2, consistency for each species was displayed using box plots that illustrated the median and interquartile ranges. However, in Figure 3, mean values of consistency for each species were displayed. To make the two figures consistent with one another, we have edited Figure 2 so that consistency is displayed as violin plots, with mean and standard error illustrated in place of median and interquartile range. Of course, at the discretion of the editor, we could return to the original Figure 2 if desired.

The two reviewers each have several comments which should improve the paper even further. My own comments are minor: First, although non-adaptive hypotheses are mentioned in the discussion, I think they deserve a bit more attention throughout the paper, possibly even in the introduction. There are many aspects of egg phenotype that seem to be relatively invariant for individual females, even when variation might be adaptive (notably, egg size). The proximate physiological mechanisms necessary to alter, say, the degree of shell pigmentation between subsequent eggs in a clutch may be formidable. Or females might be constrained in the amounts of biliverdin or other pigments, and so forth. Alternatively, there may be no selective advantage or disadvantage to consistency, so it may be essentially random. I mention this partly because it might help interpret the results, but also because it is important to keep the null hypothesis in mind. In the discussion (line 475), for example, you mention that some unparasitized species show a "surprisingly high" level of interclutch distinctiveness (surprising because they are not parasitized). But perhaps this simply indicates that there is no selection *against* distinctiveness, so it reflects variation in individual condition (or something else about individual phenotype).

- **Reply:** We thank the editor for this point. The idea that consistency may be a "default" feature of egg clutches is in line with the fact that we found no differences in consistency between parasitised and unparasitised species, suggesting that the variation we observed in consistency across species is not attributable to selection from brood parasites. Regarding individual condition, there seems to be rather little evidence that egg phenotypes are flexible in relation to individual or environmental conditions, but certainly your point is well-taken that individuals may differ non-adaptively for unmeasured reasons.
- To address this, we have added new text to the discussion which we hope highlights non-adaptive explanations that may be relevant to our results.
- The new text at lines 412-425 reads, "Which defence is elevated by coevolution with parasites may be influenced by selection from ecological factors aside from brood

parasitism: certain egg phenotypes may be costly with respect to thermoregulation, protection from UV radiation, or camouflage [54–57], and increased susceptibility to host-colonisations by other species or host-races of parasite [30,38], potentially limiting distinctiveness. Moreover, non-adaptive factors may also mean that the null hypothesis is not necessarily that both consistency and distinctiveness should be low in the absence of parasitism. For example, non-adaptive mechanisms might account for the high levels of consistency observed in some unparasitised species; we could speculate that certain egg phenotypes may be relatively invariant within females for mechanistic reasons during pigment deposition. Variation in such potential constraints may influence whether high consistency or distinctiveness is favoured, and thus why the closely-related, sympatric species studied here have taken divergent trajectories from their similar phylogenetic and ecological starting points.”

- Additionally, we have removed the text about surprisingly high levels of interclutch distinctiveness in unparasitised species, because this point is now addressed in the new text described above.

Second, I appreciated the schematic figure 1, but I think it would be even better if you had shown photos of your eggs with real examples of consistency and distinctiveness. I realize that this may be impossible/too much work, so it's only a suggestion if feasible.

- **Reply:** We thank the editor for this excellent suggestion. We have edited Figure 1 to include images of real eggs from species that align roughly with the four categories illustrated in the schematic (high absolute/high combinatorial, high absolute/low combinatorial, low absolute/high combinatorial, and low absolute/low combinatorial). These photos now appear as a separate set of panels below the schematic, and the photos are described in the revised legend for Figure 1.

Finally, this may need just a quick clarification, but I was a little concerned about the inclusion of some species that lay immaculate eggs. It seems that including these could throw off statistical analyses that are otherwise dealing with the extent of color/patterning, even if the methods themselves can theoretically accommodate such data. For example, do the immaculate eggs vary as much in background color as the patterned eggs do, or are they more uniform overall (not within vs between clutches, but at the population level)?

- **Reply:** Within the species in our dataset, there is not a one to one correspondence between background colour consistency and presence or absence of patterning. Although the immaculate species in our dataset do lay eggs that are relatively consistent in background colour (though there do exist hosts in other systems with colour polymorphisms in immaculate eggs, so this is not mechanistically impossible), some species that lay patterned eggs also exhibit no polymorphism in background colour (e.g. *Cisticola erythrops*, in which the background is consistently turquoise).
- The editor is right to point out that inclusion of these immaculate species in analyses regarding consistency could potentially affect our results. However, we believe that this issue has been addressed in our paper, in two ways. First, immaculate species were not included in any analyses that involve absolute or combinatorial distinctiveness, since distinctiveness (specifically entropy) calculations cannot accommodate immaculate species (as described in lines 229-232). Additionally, all analyses involving consistency (which can be

calculated in immaculate species) were repeated without the three immaculate species (as described in, e.g. line 309). The results from the analyses without the three immaculate species are detailed in Table S2, and in no instance did our conclusions differ between analyses with and without the immaculate species.

I enjoyed reading this exciting paper and hope that these comments are helpful.

Referee: 1

It has been hypothesized that hosts of brood parasites should minimize intraclutch egg variation (i.e., “consistency”) and maximize interclutch egg variation (“distinctiveness”) within a species to circumvent the evolution of egg mimicry by brood parasites. To date, studies investigating this possibility have lacked consensus. The authors of this manuscript investigated possible explanations for these inconsistencies by focusing on the eggs of two ancient host lineages in Africa, weavers and warblers, which are parasitized by parasites that lay mimetic eggs. In addition to testing the traditional hypothesis that both consistency and distinctiveness should be elevated, they tested the strategy-blocking hypothesis which suggests that either consistency or distinctiveness can be elevated to thwart parasitism, rather than requiring both simultaneously.

The authors found evidence that these strategies can be alternatives to combat parasitism rather requiring both to be expressed simultaneously. Contrary to expectations, combinatorial distinctiveness was more important than absolute distinctiveness. Importantly, they also found a negative relationship between consistency and absolute distinctiveness in hosts and non-hosts alike, indicating that not surprisingly, other factors affect egg appearance (e.g., Wisocki et al. 2020). This is an exciting and novel study building on the results of their previous work and I have relatively few comments.

- **Reply:** We thank the reviewer for their enthusiasm for our manuscript, and also for the thoughtful and constructive question below.

That only consistency or distinctiveness is required to combat parasitism is similar to other defenses (e.g., egg rejection and aggressive nest defense) where many hosts, especially those of the cowbird, invest most of their effort in a single defense. In the Introduction you mention Lahti’s work is one of the rare cases where we see low intraclutch and high interclutch variation in eggs of a host species. Given your findings that this combination appears to be unnecessary, why is that situation unique? If only consistency or distinctiveness is necessary, why are both present in this system? It would seem costly to invest in both if it’s not required.

- **Reply:** We thank the reviewer for this interesting and thought-provoking question. Although we can’t provide a definitive answer, we think that the scale of variation in both consistency and distinctiveness in our dataset is quite important to consider. There is no absolute value of either that we can definitively say is high or low, but rather species exhibit high or low consistency or distinctiveness relative to the other species in this dataset. As a result, it’s difficult to directly compare these results with those previously found by Lahti. However, we have produced a version of Figure 3 in which *Ploceus cucullatus* (a black square in the figures below) is labelled, and interestingly in our dataset it is roughly intermediate in both traits, perhaps implying that it is not unnecessarily extreme in either (please see the PDF copy of the response to reviewers for the figure):

Referee: 2

I found this to be a fascinating study, very well written with a clear thoughtful framework that captures the nuances associated with defensive adaptations to brood parasitism.

- **Reply:** We thank the reviewer for their very kind words about our manuscript, as well as for their constructive feedback and comments.

I only have a couple of general comments. The first is about the categorical assignment of whether a host species is parasitized or not. This is based on nest records in the area from which the clutches were collected which is great, but I would like to see more details about exactly how these categories were assigned. There were 1457 nest records for 25 species – so an average about 60 nests per species. Surely some of these species had many more nest records and some had less? Was there much variation in the rate of parasitism in the species that were identified as hosts? If so, does that provide any predictive power regarding egg signature variation? In sum, I thought this aspect was somewhat glossed over and I would certainly appreciate more details about it. Thinking of being parasitized as a yes or no trait seems slightly simplistic to me when it seems that it is more likely a continuum. Perhaps a table with breeding records per species, number of clutches parasitized and frequency of clutches parasitized would be useful? Possibly a frequency histogram of frequency of parasitism – if it is clearly bimodal then that would justify the categorization of either yes or no for species as being parasitized

- **Reply:** We thank the reviewer for this comment. The primary reason for parasitism status being treated as a categorical rather than a continuous variable in this study is that parasitism rate is not equal to parasite pressure: estimates of parasitism rate are an imperfect proxy for strength of selection. For example, a low parasitism rate may arise not because parasitism pressure is low, but because host rejection is very effective, which could in turn arise for multiple reasons (effective signatures, early stage arms race, low cost of type I errors, etc).

- We have taken several measures to account for these uncertainties in our data. First, we included eggs in our study from areas nearby to our focal area (the districts of Monze and Mazabuka), in case parasitism was occurring in nearby areas but not in Choma (as stated in lines 147-148 and 152). Additionally, we reanalysed all of our data but with species parasitised elsewhere in Africa (but not in Choma) treated as parasitised, as detailed in lines 160-163 and supplemental Tables S2-S4.
- The reviewer is absolutely correct that additional information and transparency about this aspect of the study could be useful to readers in interpreting our data. Therefore, we have added data to Table S1 which details the total number of collected clutches for each species, the number of parasitised clutches collected for each species, and parasitism rate as a percentage.
- Additionally, we have added new main text which lays out summary statistics regarding the number of collected clutches and parasitism rate, and which provides more information overall regarding how species were classified as parasitised or unparasitised. The new text at lines 150-163, reads: “To classify species as parasitised or unparasitised, we used data on 1490 collected clutches (range: 10–227 clutches, mean: 59.6 clutches) for our 25 study species, collected in the Choma region (including the Monze and Mazabuka districts) over 38 years by JFRCR and LH (Table S1). Each collected clutch was labelled as either parasitised or unparasitised, from which we calculated a parasitism rate in the Choma region (range: 2.17–42.7%, Table S1). These rates are an imperfect index of parasitism pressure (e.g. some parasitic eggs could have been rejected before nests were found), but give a reliable indication of whether a species was regularly parasitised during the study period. We therefore categorised all species with non-zero parasitism rates as ‘parasitised’. However, we cannot know with certainty whether any currently unparasitised species have previously acted as hosts. As a precaution, we therefore repeated all analyses treating as parasitised five locally unparasitised species with published parasitism records from elsewhere in their range [41] (Table S1)”

The other aspect I struggled to wrap my head around is the basic argument that that selection from brood parasites should result in higher levels of both defenses. Thus consistency and combinatorial distinctiveness as both potential adaptations against parasitism are basically predicted to be higher in parasitized species? But if either of these are effective adaptations against parasitism – shouldn't it be associated with reduced parasitism? So if you classify species as parasitized or not, I would have predicted that the parasitized species would have less consistency and combinatorial distinctiveness, thereby making them more susceptible to getting fooled by the parasites. Is the above argument a possible reason why “previous results have been inconsistent in their support for this elegant hypothesis” line 408

- **Reply:** Thank you for this interesting point. We agree that what we categorise as “unparasitised” species could be a mixture of species that haven't yet been colonised by a brood parasite, and species that have locally defeated their brood parasite thanks to their excellent defences (as your scenario above entails). One of the challenges of studying a very dynamic process like coevolution! We cannot be entirely sure at what point in time in an arms race we are sampling: some species in our dataset may have a long history parasitism, for others it may be relatively new, and still others may have locally defeated the parasite. This is an inevitable problem shared by all studies of brood

parasites (see line 426), and was the motivation behind some of the decisions we made in in our study, which we hope will help to reassure the reviewer:

- First, throughout, we repeated all of our analyses with those species currently unparasitised in Choma (but parasitised elsewhere in Africa) labelled as parasitised. Thus, if any of the species in our dataset have locally defeated the parasite, we could still potentially detect the signature of recent, intense selection pressure from brood parasites in the analyses where these species are classified as parasitised. This helps to account for the fact that some species may have previously acted as hosts, but locally defeated their parasite.
- Second, we treated parasitism as a categorical variable, which ensures that even when parasitism rate is low (which could occur because parasitism pressure is evolutionarily new, truly is infrequent, or because a species has developed advanced egg rejection behaviour), a species is still counted as experiencing parasitism pressure. We illustrate this argument with discussion of *Cisticola chiniana* (lines 429-432), a species in our dataset that is currently unparasitised at our study site but which shows intermediate levels of distinctiveness and has previously been shown to exhibit intermediate levels of discrimination behaviour (Spottiswoode and Stevens 2011).